# Young Shoots of White and Red Headed Cabbages Like Novel Sources of Glucosinolates as Well as Antioxidative Substances

**DOI:** 10.3390/antiox10081277

**Published:** 2021-08-12

**Authors:** Joanna Kapusta-Duch, Barbara Kusznierewicz

**Affiliations:** 1Department of Human Nutrition and Dietetics, Faculty of Food Technology, University of Agriculture in Krakow, 122 Balicka St., 30-149 Krakow, Poland; 2Department of Food Chemistry, Faculty of Chemistry, Technology and Biotechnology, Gdańsk University of Technology, 11/12 G. Narutowicza St., 80-233 Gdańsk, Poland; barbara.kusznierewicz@pg.edu.pl

**Keywords:** antioxidant activity, glucosinolates, young shoots, isothiocyanates, indoles, red headed cabbage, white headed cabbage

## Abstract

Most literature data indicate that the diet rich in plant products reduces the risk of developing chronic non-communicable diseases and cancer. *Brassica* vegetables are almost exclusively synthesizing glucosinolates. Glucosinolates are higher in sprouts than in mature plants, being related to the activity of the specific myrosinase involved in the degradation of glucosinolates during developmental stages. This study compares the content of total glucosinolates with their profile and, rare in the literature, also with products of their degradation. Average amounts of total glucosinolates in young shoots of white and red headed cabbage were 26.23 µmol/g d.m. and 27.93 µmol/g d.m., respectively. In addition, antioxidative properties of 21-day-old shoots of white and red headed cabbage were assessed. The area of negative peaks after post-column derivatization with the ABTS reagent, indicating antioxidant activity of young red cabbage shoots, was 20185, compared to the value determined for young white cabbage shoots (3929). The results clearly indicate that, regardless of the vegetable species, young shoots of white and red headed cabbage can be an important source of bioactive substances in the diet, thus being an important element of cancer chemoprevention.

## 1. Introduction

There is very little information in the available literature on the nutritional and non-nutritional value of young shoots of *Brassica* vegetables. Vegetable young shoots are a completely new group of food products, which is developing intensively. In germinating seeds, physiological processes, including enzymatic ones, start rapidly and the content of valuable bioactive and health-promoting components can increase several times. This refers to, for example, free amino acids, di-, tripeptides (which are important antioxidants), vitamins and polyphenols. Thus, such products are valuable additives to commonly consumed courses, often serving a decorative function as well as accompanying traditional dishes. In view of the above, they can become an invaluable source of bioactive components. Moreover, consumption of young *Brassica* shoots does not require any culinary processing; they can be grown by the consumer [1]. What is more, there are no commercially available plants at this stage of development, such as several-day-old shoots.

Numerous epidemiological studies indicate that a diet rich in plant products reduces the risk of developing chronic non-communicable diseases and cancer [2,3]. *Brassica* vegetables play a prominent role in this process, as they contain several bioactive compounds which act as antioxidants and have other health-promoting properties. The occurrence of such compounds in a food determines its functional and health-promoting nature. Therefore, *Brassica* vegetables are classified as functional foods [4,5].

*Brassica* vegetables as a group may contribute more cancer-inhibiting phytochemicals to the diet than any other group of vegetables. Bioactive compounds found in *Brassica* plants include, i.a., phenolic compounds, phytosterols, carotenoids, and—the most characteristic of this plant family—glucosinolates [5].

Polyphenols are wide group of specialized metabolites associated with the health benefits. They range from simple and single aromatic-ringed compounds (with a low molecular-weight) to large and complex tannins and derived polyphenols. Polyphenols can be classified in flavonoids (e.g., flavanols, flavones, flavan-3-ols, flavanones, isoflavones, and anthocyanidins) and non-flavonoids (e.g., phenolic acids, stilbenes, and hydroxycinnamates). They are the best studied and extensively reported group of specialized metabolites attributed to the management of among others type 2 diabetes, obesity, metabolic syndrome, atherosclerosis, neurodegenerative diseases, and cancer. The most widespread and different groups of polyphenols in *Brassica* species are the flavonoids (flavonols and anthocyanins) and the hydroxycinnamic acids. It was reported that polyphenols, together with other compounds, significantly contribute to the biological activity of *Brassica* plants [2,3].

Glucosinolates and their decomposition products (isothiocyanates, nitriles, thiocyanates, epithionitriles, and oxazolidines) show not only anticancer, but also antioxidant, anti-inflammatory, anti-allergic, anti-fungal, anti-virus, anti-mutagenic, and anti-bacterial properties. The total amount, as well as the abundance of specific glucosinolate compounds, is extremely variable between *Brassica* vegetables. Sprouts of the *Brassica* vegetables are also a rich source of glucosinolates. The consumption of broccoli and radish sprouts is currently recommended, and the fact that they are usually eaten raw makes them an excellent source of these compounds [6,7].

Numerous studies have described the glucosinolates and their breakdown products in these vegetables at full maturity. However, the literature is lacking the description of glucosinolates and their breakdown products as well as selected antioxidative substances of young shoots of white and red cabbages. These substances can be the great agents for human health.

A comparison of antioxidant activity of different products provides an important tool to determine their beneficial effects on human health. Total antioxidant activity is the result of the presence of several classes of compounds, which affect human health differently. Unfortunetly, in vitro determination does not reflect all of the biological activity. The contribution of any compounds and/or effects in vivo should be analyzed through special models that provide an image of the interaction and bioassimilation of these molecules [8]. In the paper of Dabulici et al. [8] the biological effect showed specificity based on the profile of the bioactive compounds. In aboved manuscript it has been shown that different phenolic compounds have a characteristic ability to be assimilated. For example, the high capacity of incorporation of quercetin into eukaryotic cells increased the degree of protection against oxidative stress. On the other hand, chlorogenic acid did not significantly influence endo-oxidative protection. Tha authors [8] reported that an accurate understanding of bioassimilation and bioavailability processes will improve nutraceutical formulation. The authors [8] also demonstrated that changes of the intestinal environment and gut microbiota via gut–brain axis pathway can cause neurological disorders. In their opinion remodeling the gut microbiota by various ways to maintain their balance might be a novel therapeutic strategy and might influence of the bioavibility of many antioxidative compounds [8]. In any case, their observations change the perspective of interpreting the results of in vitro tests in this area.

In *Brassica* vegetables, the most desirable glucosinolates are glucoraphanin and glucobrassicin. The highest levels of total glucosinolates were found in turnip (5.98 μg/g f.m.), cauliflower (5.49 μg/g f.m.), and white-headed cabbage (5.18 μg/g f.m.) sprouts [9]. Of glucobrassicin breakdown products, indole-3-carbinol (3-hydroxymethylindole) (I3C) is one of the most valuable and its chemopreventive action, like that of sulforaphane, is widely known and documented [10].

The sprouts of *Brassica* vegetable are considered to have greater nutritional value than the mature vegetable. They are also characterized by low calories, high concentrations, and bioavailability of micronutrients and non-nutrients and high biological activity [11]. In the available literature, there are extremely few data about the basic chemical composition of young shoots of vegetables, including white and red headed cabbage and their health-promoting properties.

In this work, apart from the antioxidant properties, the analysis was carried out not only of the changes of the sum of glucosinolates, but also of individual glucosinolates and their breakdown products. To our best knowledge there are, unfortunately, very few works which represent not only the amount of glucosinolates and their profile, but also the products of their decomposition. The novelty was also the selection of the tested material, i.e., young shoots of *Brassica* vegatbles, which are very rarely or not at all consumed by consumers. Young shoots of white and red headed cabbages are the subject which should be the aim of intensive research. This work focuses mainly on glucosinolates and antioxidants as such, the two main groups of phytochemicals responsible for the health-promoting properties of cabbage. In addition, it can be mentioned that an attempt was made to compare the profiles of these compounds between two varieties of different colors, i.e., white and red cabbage, to get the answer whether maybe one of them is for some reason better and more worth recommending. The content of glucosinolates and their breakdown products, together with the antioxidant activity of the studied young shoots of Brassica vegetables, will help to assess their health potential.

A number of reducing phytochemicals are indeed valuable bioactive compounds, so their monitoring along the food or dietary supplement production chain is increasingly recognised as an important issue, especially in the case of the so-called functional food industry. However, in this case determining the total antioxidant activity may not give a full picture of the input of individual valuable compounds. Therefore, it would be important from this point of view to use tools that would allow monitoring the profile of antioxidants. The ability to obtain such a “fingerprint” of antioxidant compounds in the plant material would provide a lot of relevant information to reveal which compounds and to what extent affect the total antioxidant potential. The online methods aim not only at the rapid measurement of antioxidative activity, but also at the profiling of individual reducing compounds in complex mixtures following their chromatographic separation from the matrix. In the most investigated approach, the solution of DPPH or ABTS radical is added post-column to the HPLC flow. Antioxidants present in a sample are detected by a decrease in absorbance at visible wavelengths due to the conversion of these radicals to their noncoloured reduced forms. Overall, this study was undertaken to broaden knowledge on health-promoting properties of raw 21-day-old shoots of white and red headed cabbages, particularly in terms of the following indicators: total and individual glucosinolates, indole-3-carbinol, indole-3-acetonitrile, 3,3′-diindolylmethane, total indoles, total isothiocyanate, and antioxidant activity.

## 2. Materials and Methods

### 2.1. Plant Material

The experimental materials were ready to eat: 21-day-old white headed cabbage shoots (*Brassica oleracea* L. var. capitata f. alba) of the cultivar *Gloria* and 21-day-old red headed cabbage shoots (*Brassica oleracea* L. var. *capitata* f. *rubra*) of the cultivar *Haco*. The seeds were purchased from one of the leading companies operating in the seed segment of the horticultural industry in Poland. The materials were cultivated in greenhouse under strictly controlled temperature conditions. The daytime temperature was 21 °C and the nighttime temperature was 18 °C. The applied peat substrate TS 2 was a mixture of specially prepared peat with the addition of PG Mix fertilizer and a surfactant, with pH = 6 and fertilization density of 2 g/L. The fertilizer used comprised of: nitrogen—14% (including: 5.5% N-NO_3_ and 8.5% N-NH_4_); phosphorus—16% P_2_O_5_; potassium—18% K_2_O; magnesium—0.8% MgO; sulphur—19% SO_3_; boron—0.03% B; copper—0.12% Cu; iron—0.09% Fe; manganese—0.16% Mn; molybdenum—0.20% Mo; and zinc—0.04% Zn.

Vegetable samples were prepared for analysis immediately after harvest. The experimental material was freeze-dried in a Christ Alpha 1–4 apparatus (Osterode am Har, Germany) and then was additionally ground in the Tecator Knifetec 1095 sample mill (Hilleroed, Denmark), until a homogeneous sample was obtained with the smallest possible particle diameter. The lyophilizates obtained were used to determine total glucosinolates and their profile, as well as glucosinolate breakdown products (i.e., isothiocyanates and indoles) and antioxidant activity. Assays were performed in three parallel replications.

### 2.2. Analitycal Methods

#### 2.2.1. Chemicals

Imidazol formate, formic acid, acetonitrile, sulfatase, 2-propanol, 2,2-azinobis-(ethyl-2,3-dihydrobenzothiazoline-6-sulphonic acid) diammonium salt (ABTS), 1,2-benzenedithiol, 3,3′-diindolylmethane (DIM), and indole-3-carbinol (I3C) were obtained from Sigma-Aldrich (Darmstadt, Germany). Glucotropaeolin (GTL) was from AppliChem (Darmstadt, Germany), indole-3-acetonitrile (I3ACN) from Merck (Darmstadt, Germany), and HPLC grade methanol from Chempur (Piekary Śląskie, Poland). Water was purified with a QPLUS185 system from Millipore (Burlington, MA, USA).

#### 2.2.2. Glucosinolates Determination

To determine content and composition of GLS in cabbage sprout samples, ISO 9167-1 method with slight modifications described earlier by Kusznierewicz, et al. [9] was used for sample preparation and HPLC analysis.

#### 2.2.3. Isothiocyanates Determination

The total isothiocyanates content in studied cabbage sprout samples was determined according to Zhang’s method [12] with modification of protocol described earlier by Pilipczuk et al. [13].

#### 2.2.4. Indole Determination

The methods of sample preparation and indole content determination used in this study were as described earlier by Pilipczuk et al. [14]. Quantification of indoles in the *Brassica* samples was performed using the standard curves generated for indole-3-acetonitrile (I3ACN), indole-3-carbinol (I3C), and diindolemethyl (DIM).

#### 2.2.5. Online Profiling of Antioxidants

Plant lyophilizates were extracted by 30 s sonication with solution containing 80% MeOH (0.05 g/mL). Next, the mixture was mixed for 30 s, again sonicated and mixed, then centrifuged for 5 min (5000× *g* at 4 °C). Finally, the obtained supernatant was collected, and the steps were repeated two times, and the extracts were combined. To obtain profiles of antioxidants, the HPLC-DAD system (Agilent Technologies, Wilmington, DE, USA) was connected with a Pinnacle PCX Derivatization Instrument (Pickering Laboratories Inc., Mountain View, CA, USA) and a UV–VIS detector (Agilent Technologies, Wilmington, DE, USA). The conditions of chromatographic separation of bioactive compounds present in plant extracts were as follows: column Phenomenex Kinetex XB-C18 100A column (150 × 4.6 mm, 5 µm); mobile phase: A—4.8% of formic acid, B—methanol; elution program: 0 min—5% B; 20 min—50% B; 25 min—100% B; 30 min—100%; flow rate—0.8 mL/min; injection volume—10 μL. The post-column derivatization procedure was carried out according to Kusznierewicz et al. [15]. The groups of phytochemicals were identified on the basis of UV-Vis spectra and classified as phenolic acids, anthocyanins, and polar fraction with unidentified compounds. The antioxidant activity of identified group of phytochemicals was assessed as a sum of area under the negative peaks recorded in HPLC analyses during post-column derivatisation with ABTS reagent.

### 2.3. Statistical Analysis

All analyses were carried out in three parallel replications and mean ± SD were calculated for the values obtained. Using one-way analysis of variance (ANOVA), the significance of differences was checked between mean values of young shoots of white and red cabbage. The significance of differences was estimated with Duncan test at the critical significance level of *p* ≤ 0.05. The Statistica 10.1 (StatSoft, Inc., Tulsa, OK, USA) program was applied. The composition of glucosinolates was expressed as µmol/1 g dry matter.

## 3. Results

At present, more than 300 different compounds, belonging to glucosinolates, have been identified. They have a similar structure based on a β-d-thioglucose group, a sulfonated oxime group, and a side chain derived from one of the seven protein amino acids. Glucosinolates are synthesized from aliphatic (leucine, isoleucine, valine, methionine, alanine), aromatic (phenylalanine, tyrosine), and indole (tryptophan) amino acids [16].

The following aliphatic glucosinolates were identified in the examined young shoots of white and red headed cabbage: glucoiberin (GIB), progoitrin (PRO/e-PRO), glucoraphanin (GRA), sinigrin (SIN), gluconapin (GNA), as well as the indolic ones: glucobrassicin (GBS), 4-hydroxyglucobrassicin (40HGBS), methoxyglucobrassicin (mGBS), and neoglucobrassicin (neoGBS) (Table 1).

The glucosinolates group compounds show high chemical stability and formation of bioactive indoles and isothiocyanates results from enzymatic hydrolysis of glucosinolates. The enzyme catalyzing this reaction is β-thioglucosidase, thioglucoside glucoside glucohydrolase EC 3.2.3.1. (myrosinase), which occurs in myosin cells, from where is released under appropriate conditions. In *Brassicas*, the degradation of glucosinolates produces both stable (allyl and 2-phenylethyl isothiocyanates) and unstable products (e.g., indole-3-carbinol) [17].

As representative samples were prepared for freeze-drying of the examined vegetables and milling was applied in pre-processing, young shoots were also examined for GLS breakdown products. These were namely: indole-3-carbinol (I3C), indole-3-acetic acid (I3AA), indole-3-acetonitrile (I3ACN), diindolylmethane (DIM), and a sum of isothiocyanates (ITC).

### 3.1. Total Glucosinolates

There were statistically significantly differences in total glucosinolates contents (*p* ≤ 0.05) between individual species of 21-day-old shoots. Average amounts of total glucosinolates in young shoots of white and red headed cabbage were 26.23 µmol/g d.m. and 27.93 µmol/1 g d.m., respectively. Young shoots of red headed cabbage contained significantly more (*p* ≤ 0.05) total glucosinolates compared to young shoots of white headed cabbage (Table 1, Figure 1 and Figure 2).

In total glucosinolates content of young shoots of both red- and white headed cabbage, the share of aliphatic glucosinolates was highest and was 23.42 and 24.96 µmol/1 g d.m., respectively. Then were indolic ones (2.81 and 2.98 µmol/1 g d.m., respectively). There were no aryl glucosinolates. Young shoots of red headed cabbage contained significantly more (*p* ≤ 0.05) aliphatic glucosinolates than those of white headed cabbage. The content of indole glucosinolates was similar in both species of young shoots (*p* > 0.05).

### 3.2. Aliphatic and Indole Glucosinolates

The content of aliphatic glucosinolates in the total amount of these compounds was the highest and amounted to 89.3% in young shoots of white cabbage and 89.4% in young shoots of red cabbage. The percentages of indole glucosinolates, similarly to aliphatic glucosinolates, were almost identical and were 10.7% (young white cabbage shoots) and 10.6% (young red cabbage shoots) (Table 1, Figure 1).

In absolute values, the highest amounts of aliphatic and indolic glucosinolates had young shoots of red cabbage (24.96 and 2.98 µmol/g d.m. of the raw material, respectively), followed by young shoots of white cabbage (23.42 and 2.81 µmol/g d.m., respectively) (Table 1). Young red cabbage shoots contained significantly more (*p* ≤ 0.05) aliphatic glucosinolates than young shoots of white cabbage.

The examined shoots of young white cabbage were found to have the highest amount of sinigrin (SIN) (60.4%), followed by progoitrin (PRO/e-PRO) (28.6%) and glucoiberin (GIB) (5.6% of total aliphatic glucosinolates content). Similar results were obtained for young shoots of red headed cabbage, in which sinigrin (SIN) comprised 50.6%, progoitrin (PRO/e-PRO) 21.8%, and glucoiberin (GIB) 12.8% of total aliphatic glucosinolates content (Table 1). As for the aliphatic glucosinolates, both examined species had the lowest amounts of glucoraphanin (GRA), being 2.7% in young white cabbage shoots and 3.6% in young red cabbage shoots. In absolute values, a similar amount (*p* > 0.05) of sinigrin (SIN) was determined in these shoots (~14.04 µmol/g d.m.). In turn, young shoots of white cabbage had a significantly (*p* ≤ 0.05) higher amount of progoitrin (PRO/e-PRO) (6.71 µmol/g d.m.) compared to young red cabbage shoots, as opposed to gibberin (GIB), of which the content was significantly (*p* ≤ 0.05) higher in young shoots of red cabbage (3.20 µmol/g d.m.).

The second group of hydrolysis products with anticancer properties, glucobrassicins (GBS), account for a significant proportion of the indole glucosinolates present in *Brassica* vegetables. Among indole glucosinolates the highest glucobrassicins proportion in this group was found in young white cabbage shoots (37.0%); almost half the amount of this compound was in young shoots of red cabbage (17.8%). The latter product, however, had significantly (*p* ≤ 0.05) more 4-hydroxyglucobrassicin compared to young shoots of white cabbage. Of all indole glucosinolates, there was the least of neoglucobrassicin (neoGBS) in both examined species of young shoots.

### 3.3. Isothiocyanates and Indoles

The content of isothiocyanates was significantly higher (*p* ≤ 0.05) in young red cabbage shoots (the absolute content of 1.55 µmol/g d.m.) then in young shoots of white cabbage (1.02 µmol/g d.m.). Regarding the sum of indoles, significantly more (*p* ≤ 0.05) of these components was in the young shoots of white cabbage (0.18 µmol/g d.m.), i.e., by 42%, compared to young shoots of red headed cabbage (0.08 µmol/g d.m.). Young white cabbage shoots were also characterized by a significantly higher (*p* ≤ 0.05) content of indole-3-carbinol (0.01 µmol/g d.m.) compared to those obtained from red cabbage (0.007 µmol/g d.m.). As for the content of indole-3-acetonitrile (I3ACN), young shoots of red cabbage contained significantly more (*p* ≤ 0.05) (by 45%) of this compound compared to respective shoots of white cabbage (Table 2, Figure 2).

### 3.4. Antioxidant Activity

Antioxidant activity of young shoots of white and red cabbage was presented as an HPLC profile, by indicating particular classes of antioxidants and defining their contribution to the total antioxidant potential of the above plants. The antioxidant activity of samples was assessed as a sum of area under the negative peaks of identified group of phytochemicals recorded during post-column derivatisation with ABTS reagent (Figure 3).

The upper panel of chromatograms in Figure 2 shows the typical phenolics profile obtained for cabbage extracts recorded before derivatization at 270 nm. Based on the UV-Vis spectra, the extracts showed mainly the presence of phenolic acids and anthocyanins. These profiles were respectively set with the chromatograms recorded at 734 nm after post-column derivatization with the ABTS reagent (lower panel of chromatograms). The presence of antioxidants in the eluate causes decrease in absorbance of ABTS reagent due to the conversion of this radical to their colorless reduced form, which results in a lowering of the baseline of the chromatogram visible as negative peaks. Such a combination of chromatograms before and after derivatization allows to indicate which of the separated analytes and to what extent affects the antioxidant activity of the extracts. Additionally, the antioxidant activity of samples was assessed as a sum of area under the negative peaks of identified group of phytochemicals recorded during post-column derivatisation with ABTS reagent (Figure 3, bar graphs) [15].

The chromatograph obtained showed that the young shoots of red cabbage had stronger antioxidant potential. Certainly, these results were largely influenced by more than five times higher content of anthocyanins, in which red cabbage is rich, even at such an early stage of development. The area of negative peaks after post-column derivatization with the ABTS reagent, indicating antioxidant activity of young red cabbage shoots was 20185, compared to the value determined for young white cabbage shoots (3929).

## 4. Discussion

### 4.1. Total Glucosinolates

The natural glucosinolates content in vegetables varies in accordance with genotype, plant developmental step, soil, and cultivation conditions and other ecophysiological influences, and also by storage conditions. Additionally, glucosinolates level vary over the growth season and several studies reported higher glucosinolates levels in *Brassica oleracea* plants grown in spring compared to plants grown in autumn [18].

In an interesting study by Kołodziejski et al. [19] the authors focused on plant material obtained from different individual parts or stages of growth of *Brassica* species. These authors proved that there are significant differences in glucosinolates composition and their conversion rate to bioactive derivatives within not only one species, but even the same plant. Kołodziejski et al. [19] reported that the highest content of total glucosinolates was determined in sprouts of white cabbage (3.57 µmol/g d.m.) and seeds (2.00 µmol/g d.m.); then respectively in stump, leaves, and roots.

Most literature data concern mature *Brassica* vegetables or sprouts, but not plants in other vegetative stages, such as young shoots. The results obtained by Ciska et al. [20] for the total content of glucosinolates in different red cabbages at full maturity depending on the year of harvest. Different results were attributed to climatic factors like rainfall and temperature during the growing period. Hassini et al. [21] reported that total glucosinolates content depended significantly on the variety, being higher in red than in white cabbage sprouts. In a study by Wermter et al. [18] red cabbage heads at full maturity produced higher levels of glucosinolates compared to white cabbage heads. According to Bell and Wagstaff [22] the white headed cabbage and red headed cabbage are widely consumed and studied crops but has modest total glucosinolates concentrations compared to those of other *Brassica* crops. In turn, studies by Drozdowska et al. [23] on young 14-day shoots of red headed cabbage demonstrated about four times lower total GLS contents (7.27 µmol/1 g d.m.) than the values obtained in this work. These results indicate that intense transformations are taking place in the growing plant and prove that 21-day-old shoots are an excellent source of these compounds: better than 14-day-old shoots of the same species and cultivar.

### 4.2. Aliphatic and Indole Glucosinolates

In only one study found on this subject concerning 14-day-old shoots of red cabbage, the main glucosinolates were aliphatic components such as progoitrin, sinigrin, and glucoraphanin which occurred at concentrations above 5.66 µmol/g d.m. (78% of total glucosinolates) [23]. Seeds have the largest amount of these reserve metabolites, and this is consistent with their roles in plant defense and survival. It has also been a consequence of glucosinolates metabolism and dilution of their concentration during tissue expansion [21]. In a study of Drozdowska et al. [23] other aliphatic compounds in glycolysis were present. According to Drozdowska et al. [23] the indolyl components such as glucobrassicin (GBS), metoxybrassicin (metoxyGBS), and neo-glucobrassicin (neoGBS) were at an average concentration of 0.87 µmol/g d.w. (12% of total glucosinolates). These results differ slightly from the results obtained in this study, because the authors do not mention the presence of another indole glucosinolate, i.e., 4-hydroxyglucobrassicin (40HGBS). Hassini et al. [21] reported that in the red variety the aliphatic glucosinolates content was 6.7-fold higher than in the white one. The authors reported also that slight differences in the indole glucosinolates content were found in both varieties. Other authors reported a higher level of total glucosinolates in *Brassica* sprouts (ready for harvest, between the 4th and 12th day after sowing) as compared to our results for 21-day-old shoots [24,25,26].

In research of Drozdowska et al. [23], the main glucosinolate in red headed cabbage were aliphatic compounds—sinigrin, glucoiberin, and progoitrin, which occurred at concentrations above 3.38 µmol/g d.m. (75% of total glucosinolates). Meyer and Adam [27] showed results concerning the main group of glucosinolates in red headed cabbage at full maturity, which were glucoraphanin, sinigrin, and progoitrin. Sasaki et al. [28] reported that the glucoraphanin content in white head cabbage leaves from 32 different cultivars was within a wide range from 106.2 to 153.9 mg/100 g of fresh weight. In their research the indole components represented a minor group like in this study.

Wermter et al. [18] monitored the variation of glucosinolates levels and the formation of their bioactive hydrolysis products in commercial red and white cabbage heads obtained from three local retailers in Germany and linked the data with the cultivation practices and post-harvest storage conditions. These authors reported that in the heads of the analyzed white and red cabbage cultivars, a total of 12 chemically different glucosinolates were detected. The main glucosinolates were allyl glucosinolate (Allyl), 3-butenyl glucosinolate (3But), 2-hydroxy-3-butenyl glucosinolate (2OH3But), 3-(methylsulfinyl) propyl glucosinolate (3MSOP), 4-(methylsulfinyl) butyl glucosinolate (4MSOB), and indol-3-ylmethyl glucosinolate (I3M). In a study of Wermter et al. [18] the two major glucosinolates detected in white cabbage at full maturity were 3-(methylsulfinyl) propyl glucosinolate and allyl glucosinolate, while red cabbage was often the richest in -hydroxy-3-butenyl glucosinolate and 4-(methylsulfinyl)butyl glucosinolate. Bell and Wagstaff [22] concluded that red cabbage contains similar total glucosinolates content to white cabbages but differs in the relative amounts present within leaf tissues, e.g., it contains greater concentrations of glucoraphanin and gluconapin and less sinigrin.

### 4.3. Isothiocyanates and Indoles

Hanschen and Schreiner [29] reported that in humans, isothiocyanates have anti-carcinogenic properties and exert a multitude of health beneficial effects, including anti-microbial, anti-inflammatory, and anti-thrombotic effects. On the other hand, nitriles and epithionitriles were shown to have less health protective potential

Conversion of glucosinolates to their bioactive derivatives takes place due to the action of myrosinase (thioglucoside glucohydrolase EC 3.2.3.1), an enzyme occurring in cells (so-called protein bodies—myrosin granules) [30]. As a result, β-thioglycosidic bond of glucosinolates is hydrolyzed with the detachment of glucose molecule and final formation of thiocyanates, isothiocyanates (solution pH below 4), sulfates, and nitriles (solution pH 4). In cells, myrosinase and glucosinolates are physically separated; therefore, for the enzymatic conversion of glucosinolates to occur, the cell must be damaged. According to researchers, the anti-cancer effect (referring mainly to sulforaphane and indole-3-carbinol) results from the effects of these compounds on phase I and II enzymes [31,32,33]. Phase II enzymes, present in the small intestinal mucosa, liver, and colon, are closely associated with the reactions of neutralizing carcinogens and initiating apoptosis of cancer cells [34,35]. Unfortunately, the conversion of glucosinolates into their health-promoting derivatives does not take place with 100% efficiency. In the case of glucoraphanin (broccoli), as much as 80% is converted to nitriles, the remaining to the pro-health sulforaphane. Scientists indicate that in human organisms the glucosinolates transformation into health-promoting derivatives can also take place under the influence of the intestinal microflora [36].

In this work some differences can be noticed in the activity of myrosinase, because in red headed cabbage more isothiocyanates were formed, i.e., mainly aliphatic glucosinolates degraded, and in white headed cabbage there was a higher degree of indole glucosinolate conversion to indoles.

In a study by Kołodziejski et al. [19], the total concentrations of main degradation products—isothiocyanates, indoles, nitriles, and epithionitriles—were determined for the investigated plant material obtained from individual parts or stages of growth for four representative crops from *Brassica* family, including white headed cabbage. The authors showed that the highest content of total isothiocyantes was determined in roots of white cabbage (5.51 µmol/g d.w.), then in leaves (4.88 µmol/g d.w.); next in seeds, stump, and finally sprouts (0.25 µmol/g d.w.). They found 0.07 µmol/g d.w. of total indoles in sprouts of white cabbage. This is not in line with the results obtained in this work. These authors concluded also that the conversion rate of glucosinolates to isothiocyhanates and indoles varied significantly between individual parts of investigated *Brassica* plants.

In a study by Wermter at al. [18], glucosinolates and glucosinolate hydrolysis product profiles in red and white cabbages from three different food retailers were monitored over six different sampling dates. In above work was shown that, glucosinolate hydrolysis product profiles and hydrolysis behavior varied considerably over the season. The highest total isothiocyanate concentrations were observed in conventional red (1.66 μmol/g f.m.) and organic white (0.93 μmol/g f.m.) cabbages. The red cabbage with up to 1.06 μmol/g f.m. of 4-(methylsulfinyl)butyl isothiocyanate (sulforaphane), was an excellent source for this health-promoting isothiocyanate.

The total isothiocynates concentration of *Brassica* vegetables, including leaves of white and red head cabbages, throughout the year was reported by Fusari et al. [37]. These authors obtained total isothiocynates concentration on the level of 38.37 (leaves of white head cabbage) and 99.9 μmol % g d.w. (leaves of red headed cabbage).

Wang et al. [38] obtained the results of isothiocyanates in white headed cabbage and red headed cabbage at full maturity in the range 40.2–67.1 and 1.3–3.8 μmol/100 g f.m., respectively. Comparative evaluation of total isothiocyanates of white cabbages at full maturity from Croatia’s regions made the team of Lončarić [39], which concluded that “*Čepinski*” cabbage showed to contain the highest amount of d-limone (40.75 µg/L) and allyl isothiocyanate (1090.26 µg/L), the most important volatile compounds responsible for the fresh cabbage flavour.

Stable degradation products include allyl and 2-phenylethyl isothiocyanates. The most important unstable product is indole-3-carbinol (I3C), a compound with well-documented effects in cancer prevention. Under hydrolysis, glucoraphanin is converted to sulforaphane or sulforaphane nitrile. Synigrin turns into allyl isothiocyanate. The product of GBS hydrolysis is indole-3-carbinol, while phenylethyl isothiocyanate is formed from gluconasturcine [10].

In a study by Hanschen and Schreiner [29] *Brasicca* sprouts often contained more than 10 times more glucosinolates or their hydrolysis products compared to fully developed vegetables. Moreover, during head development, both glucosinolate concentrations as well as hydrolysis product concentrations changed, and mini heads contained the highest isothiocyanate concentrations. The authors concluded that from a cancer-preventive point of view, consumption of mini heads of the *Brassica oleracea* varieties is recommended.

### 4.4. Antioxidant Activity

In general, the phenolic composition and the antioxidant activity of vegetables are known to vary depending on the cultivar, the environmental conditions (including temperature, humidity, and UV irradiation) at the site of collection and the growing stage [40].

In a study by Li et al. [41], anthocyanin levels were higher in seedlings grown under light conditions compared to those grown under dark conditions for both –white and red head cabbage cultivars. In above study red cabbage accumulated 1.94–4.05 times greater total anthocyanin in 4-, 6-, 8-, and 10-day-old seedlings, when compared to white cabbage cultivar under light/dark and dark conditions.

In a study by Drozdowska et al. [42], the antioxidant activity of plant samples was determined using three methods: with ABTS˙^+^, DPPH and FRAP reagents. The applied methods were adequate for the examined material—young shoots of red headed cabbage showed a higher radical scavenging activity in comparison to white headed headed cabbage. Baenas et al. [26], who studied 4-day-old red cabbage sprouts, demonstrated a similar radical scavenging activity in DPPH and FRAP assays. Seven-day-old red cabbage sprouts showed a high antioxidant activity in FRAP and DPPH assay, but the results were lower in comparison to Drozdowska et al. [42] data [43].

The available literature presents the results of antioxidant activity, expressed as μmol Trolox per g of fresh vegetables, but in mature white and red headed cabbage. The values and ranges reported by various authors in mature white cabbage differ and are as follows: 3.3–4.9, by Kusznierewicz et al. [44]; 1.3–1.8 μmol Trolox/g fresh vegetables by Podsędek et al. [45]. However, the values determined by Wu et al. [46] and Ou et al. [47] were much higher, amounting to 13.6 μmol Trolox/g of fresh vegetables and 23.0–146.0 μmol Trolox/g dry matter, respectively. Regarding antioxidant activity of mature red headed cabbage, the values provided by various authors are: 9.8–12.6, by Podsędek et al. [45]; 22.5, by Wu et al. [46]; and 9.2–34.0 μmol Trolox/g of fresh vegetables, by Leja et al. [48].

The discrepancies between the quantity of biologically active compounds and variation in the antioxidant activity of vegetable samples studied and that reported by other authors can be caused by many factors. The differences between content of vitamin C, carotenoids, and polyphenols might result from natural variation of tested material, time of harvest and ripening state, cultivation, environmental conditions, and condition of post-harvest storage [49].

Chemical methods for the determination of antioxidant capacity are based on measurement of the effect of antioxidants on the rate of oxidation processes occurring in the sample (ORAC and TRAP), reduction of metal ions, e.g., iron (FRAP) or copper (CUPRAC), synthetic radical scavenging capacity (ABTS, DPPH) or measuring the amount of lipid oxidation products or LDL fractions. It is noteworthy that despite there being a number of methods for the assessment of antioxidant properties, there is no standardization of the results obtained. Many variants of a given method are often known, which differ in measurement conditions to such extent that comparison of the results is impossible. The method using the HP-LC suite may seem to be devoid of these many drawbacks, but it is not perfect [50].

Results of Drozdowska et al.’s [42] research indicated that antioxidant activity of red cabbage decreases with time of cultivation. This suggests that red cabbage could have the highest antioxidant activity when the leaves are young.

## 5. Conclusions

The content of total glucosinolates and aliphatic glucosinolates in 21-day-old shoots of red cabbage was significantly higher, compared to young shoots of white cabbage. The amount of active anticancer compounds, i.e., sinigrin, in young shoots was close, regardless of the species, while glucoraphanin content was significantly higher (by 29%) in young shoots of red cabbage compared to shoots of white cabbage. In addition, when compared to red headed cabbage shoots, young shoots of white headed cabbage contained significantly more progoitrin, the substance unfavourable in view of thyroid functioning. Total isothiocyanate content was significantly higher in young shoots of red headed cabbage compared to young shoots of white headed cabbage. An inverse relationship was observed for all indole products of glucosinolates decomposition, of which the amount was significantly larger in young shoots of white cabbage. Anthocyanins were the component responsible for stronger antioxidant activity of young shoots of red headed cabbage in comparison to young shoots of white headed cabbage. Young shoots of *Brassica* vegetables, freshly harvested and added to dishes, can provide an excellent and pure source of bioactive substances as well as possibly reducing the risk of chronic non-communicable diseases and cancer. The total antioxidant activity is the result of the presence of several classes of substances, which affect human health differently. The determination of bioavailability is much more significant in vivo as an indicator of cell absorption. This aspect should be confirmed by further experiments.

## Figures and Tables

**Figure 1 antioxidants-10-01277-f001:**
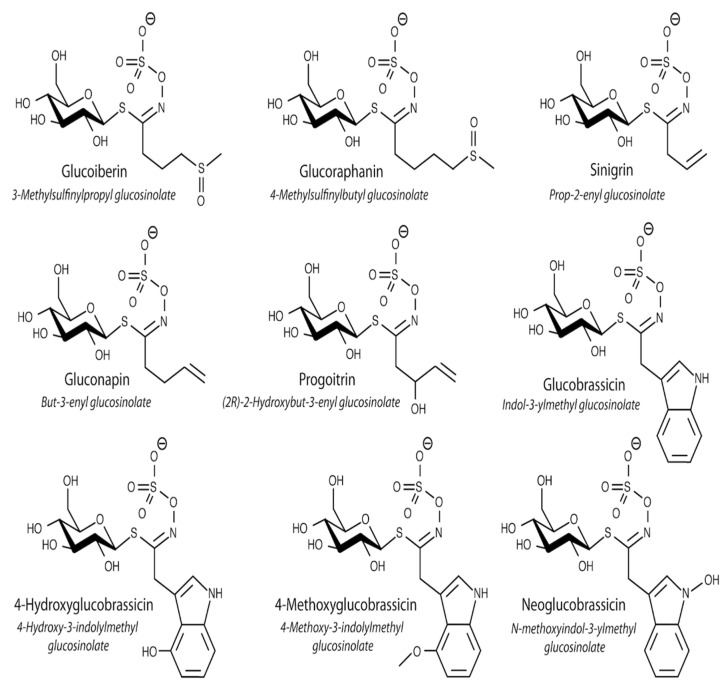
The structures of glucosinolates detected in young shoots of white and red cabbage.

**Figure 2 antioxidants-10-01277-f002:**
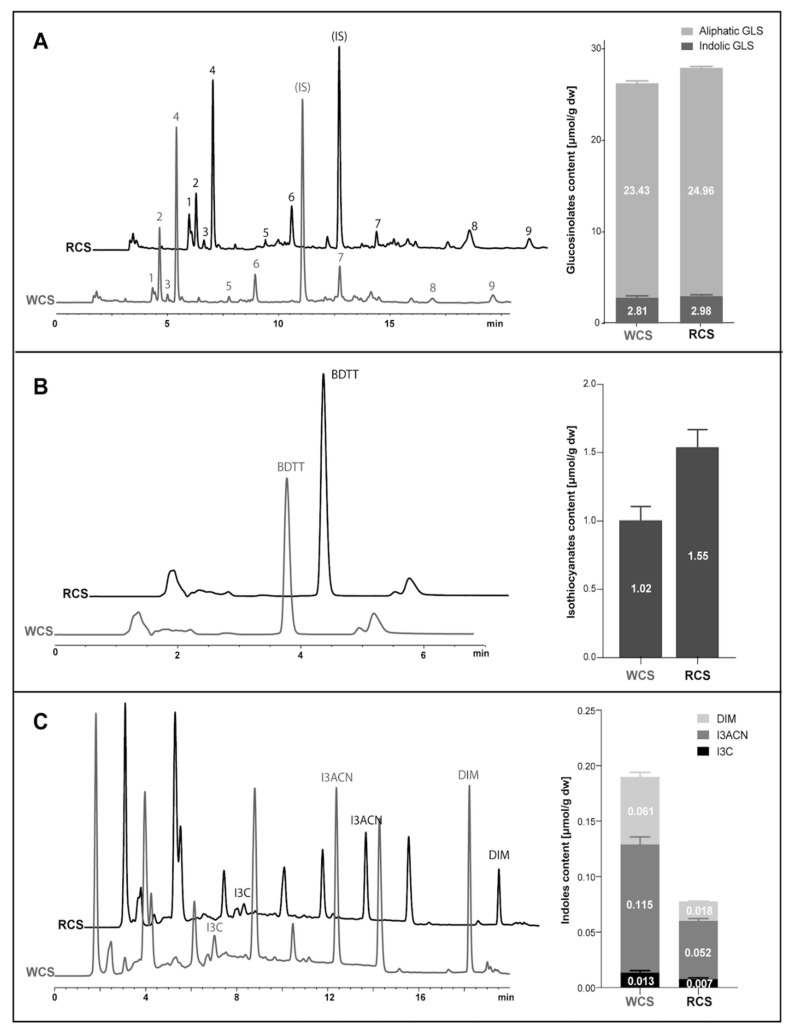
The chromatograms of glucosinolates (**A**) and products of their autolysis: isothiocyanates (**B**) and indoles (**C**) obtained during HPLC analyses of extracts obtained from white and red cabbage shoots (WCS and RCS, respectively). The glucosinolates were determined as desulfo- derivatives and chromatograms were recorded at 229 nm. The names of glucosinolate peaks are given in Table 1. The total isothiocyanates content were determined after cyclocondensation reaction with 1,2-benzenodithiol to 1,3-benzeneditiole-2-thione (BDTT) that peaks were registered at 365 nm. The detection of indolic products of degradation of glucosinolates was performed with the use of fluorescence detector (ex./em.—280/360 nm). Abbreviation used: IS, internal standard; I3C, indole-3-carbinol; I3ACN, indole-3-acetonitrile; DIM, diindolylmethane.

**Figure 3 antioxidants-10-01277-f003:**
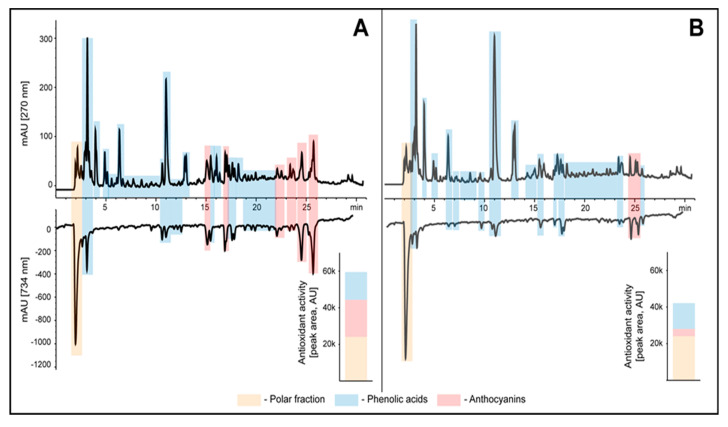
Sample HPLC chromatogram of extracts from red cabbage young shoots (**A**) and white cabbage young shoots (**B**) (top chromatograms at 270 nm) along with profiles of antioxidants detected online with ABTS reagent (bottom chromatograms at 734 nm). The groups of phytochemicals were identified on the basis of UV-Vis spectra. The antioxidant activity of samples was assessed as a sum of area under the negative peaks of identified group of phytochemicals recorded during post-column derivatisation with ABTS reagent and presented as a bar graphs.

**Table 1 antioxidants-10-01277-t001:** Content of glucosinolates in young shoots of white cabbage and young shoots of red cabbage (µmol/g d.m.).

Peak No	Glucosinolate	Young Shoots of White Cabbage	Young Shoots of Red Cabbage
1	Glucoiberin	1.31 ± 0.14 ^a^	3.20 ± 0.16 ^b^
2	Progoitrin	6.71 ± 0.29 ^b^	5.43 ± 0.06 ^a^
3	Glucoraphanin	0.64 ± 0.10 ^a^	0. 90 ± 0.04 ^b^
4	Sinigrin	14.15 ± 0.33 ^a^	13.93 ± 0.37 ^a^
5	Gluconapin	0.61 ± 0.02 ^a^	1.50 ± 0.06 ^b^
6	Glucobrassicin	1.04 ± 0.04 ^a^	0.53 ± 0.09 ^b^
7	4-hydroxyglucobrassicin	0.80 ± 0.13 ^a^	1.47 ± 0.10 ^b^
8	Metoxyglucobrassicin	0.64 ± 0.10 ^a^	0.58 ± 0.08 ^a^
9	Neoglucobrassicin	0.32 ± 0.02 ^a^	0.39 ± 0.09 ^a^
Sum of aliphatic glucosinolates		23.43 ± 0.28 ^a^	24.96 ± 0.16 ^b^
Sum of indole glucosinolates		2.81 ± 0.21 ^a^	2.98 ± 0.16 ^a^
Total gucosinolates		26.23 ± 0.39 ^a^	27.93 ± 0.32 ^b^

Values are presented as mean value ± SD (*n* = 3) and expressed in dry matter. Means in rows with different superscript letters in common differ significantly (*p* ≤ 0.05).

**Table 2 antioxidants-10-01277-t002:** Content of decomposition products in young shoots of white cabbage and young shoots of red cabbage (µmol/g d.m.).

Compounds.	Young Shoots of White Cabbage	Young Shoots of Red Cabbage
Indole-3-carbinol	0.01 ± 0.00 ^b^	0.00 ± 0.00 ^a^
Indole-3acetonitrile	0.11 ± 0.01 ^b^	0.05 ± 0.00 ^a^
Diindolylmethane	0.06 ± 0.00 ^b^	0.02 ± 0.00 ^a^
Total indoles	0.19 ± 0.01 ^b^	0.08 ± 0.00 ^a^
Total isothiocyanates	1.02 ± 0.08 ^a^	1.55 ± 0.13 ^b^

Values are presented as mean value ± SD (*n* = 3) and expressed in dry matter. Means in rows with different superscript letters in common differ significantly (*p* ≤ 0.05).

## Data Availability

Data is contained within the article.

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
