# Peer review of "Young Shoots of White and Red Headed Cabbages Like Novel Sources of Glucosinolates as Well as Antioxidative Substances"

_antioxidants, 2021, doi:10.3390/antiox10081277_

Round 1
Reviewer 1 Report
Dear Authors,
After the review process, I have several comments: you should include numerical data in the abstract; you should clearly define the control in the case of results from section 3.4; you should expand the information about bioavailability, because (page 2, line 59-64) the full affirmations are not correct. You could find newly published data that presents the bioactive potential of functional products and the bioavailability of phenolic compounds. In addition, because part of these compounds (identified by you) you should present more comments about a link between microbiota dysbiosis, and neurodegenerative pathogenesis; the number of references from the introduction is too little for an article, and for these studies; you should rewrite the conclusion section, and include a mention about future studies and alternative valorization of the results.
Author Response
Dear Reviewer,
please see the attachment.
The authors are very grateful to the anonymous Referees for their valuable comments.

Reviewer 2 Report
The topic of this study is of interest. Yet I recommend the authors to address the following comments:
1) The title is too general and does not reflect the study.
2) Keywords: replace Brassica (already in the title) by the specific Brasssica varieties used.
3) In the introduction, authors refer to the several bioactive molecules of Brassica…but then, they do not clearly state the reason for just analysing the levels of glucosinolates. Then again, they opted for looking at antioxidant activity? And associated with phenolic compounds. Again: what is the message that they want to pass to the readers?
4) Please make correlation (numbers) between table 1 and Figure 1.
5) I suggest that authors insert a Figure with representative structures of the glucosinolates classes.
6) 2.2.5. On-line profiling of antioxidants. Explain the use of 270 nm. Explain the peaks in chromatograms at 734 nm. What is expected from a control (absence of extract/antioxidant)?
Author Response

(The authors gave the same response as above.)

Reviewer 3 Report
In this manuscript Kapusta-Duch et al., studied. young shoots of selected Brassica vegetables possess more bioactive substances suggesting they are more important for human health (e.g. anti-cancer). The work is interesting.
Under 3.1. Total glucosinolates (Table 1), the average amounts of total glucosinolates in young shoots of white and red headed cabbage were 26.23 μmol/g d.m. and 27.93 μmol/1 g d.m., These levels are very close (only 6% difference) although they are statistically significant.
4.4. Antioxidant activity. As you discussed in this section that the phenolic compounds are major contributor to this direct antioxidant activity. You could have quantified the different polyphenols in these young shoots of white and red cabbages. On the other hand, glucosinolates/isothiocyanates are not direct antioxidants, they may reduce the risk of chronic non-communicable diseases and cancer via different mechanisms. One assay on Nrf2 or phase 2 antioxidant enzyme expression in cultured cells could be performed to provide more solid chemoprevention evidence on these young shoots of selected Brassica vegetables (in comparison with their aged controls).
Author Response

(The authors gave the same response as above.)

Round 2
Reviewer 1 Report
Dear Authors,
No other comments. Best regards.
Reviewer 2 Report
The authors addressed some of my major comments, whereas there are still some important corrections to be made:
1) The added text in introduction (justifying glucosinolates) is not adjusted to the context of the manuscript. Note that lines 49-50 are about Brassica; then lines 50-53 are about glucosinolates…then again line 54 returns to Brassica… please rephrase all this part in order to make sense (possibly delete lines 49-54)
2) Why authors do not highlight the relevance of phenolic compounds also in introduction, together with glucosinolates? This “issue” still sounds confusing…
3) Regarding “Please make correlation (numbers) between table 1 and Figure 1.” – please add a column with numbers in Table 1, for each compound. Then, please used the same number to identify the respective peak chromatogram in Figure 1 (old version).
4) Regarding “I suggest that authors insert a Figure with representative structures of the glucosinolates classes”- My suggestion was to “draw chemical structures” Instead, the authors introduced one Figure with the same values of Table 1. Please delete that figure since it does not add any new data.
Round 3
Reviewer 2 Report
The authors have addressed all my comments